The impact of COVID-19 and control measures on public health in Thailand, 2020

Yorsaeng Ritthideach 1
Suntronwong Nungruthai 1
Thongpan Ilada 1
Chuchaona Watchaporn 1
Lestari Fajar Budi 1
Pasittungkul Siripat 1
Puenpa Jiratchaya 1
Atsawawaranunt Kamolthip 2 3
Sharma Chollasap 3 4
Sudhinaraset Natthinee 1
Mungaomklang Anek 2
Kitphati Rungrueng 5
Wanlapakorn Nasamon 1
Poovorawan Yong Yong.P@chula.ac.th 1
1 Department of Pediatrics, Faculty of Medicine, Chulalongkorn University, Center of Excellence in Clinical Virology , Pathumwan , Thailand
2 Department of Disease Control, Ministry of Public Health, Institute for Urban Disease Control and Prevention , Bangkok , Thailand
3 Department of Disease Control, Ministry of Public Health, Institute of Preventive Medicine , Nonthaburi , Thailand
4 Department of Disease Control, Ministry of Public Health, Division of International Disease Control Port , Nonthaburi , Thailand
5 Ministry of Public Health , Nonthaburi , Thailand
Kabir Russell
Electronic publication date: 2022 Feb 16
Publication date: 2022
Volume: 10
Electronic Location ID: e12960
Received 2021 Aug 27; Accepted 2022 Jan 27
Copyright: ©2022 Yorsaeng et al.
Copyright year: 2022
Copyright holder: Yorsaeng et al.
License: This is an open access article distributed under the terms of the Creative Commons Attribution License, which permits unrestricted use, distribution, reproduction and adaptation in any medium and for any purpose provided that it is properly attributed. For attribution, the original author(s), title, publication source (PeerJ) and either DOI or URL of the article must be cited.
License URL: https://creativecommons.org/licenses/by/4.0/

Keywords: COVID-19, Public health, Mental health, Impact, Viral disease, Road accident, Suicidal behavior

Funding: The MK Restaurant Company Limited The Ratchadaphiseksomphot Endowment Fund The Center of Excellence in Clinical Virology of Chulalongkorn University/King Chulalongkorn Memorial Hospital GCE 59-009-30-005 This work was supported by the MK Restaurant Company Limited for Ritthideach Yorsaeng, the Ratchadaphiseksomphot Endowment Fund for a Postdoctoral Fellowship for Jiratchaya Puenpa and the Center of Excellence in Clinical Virology of Chulalongkorn University/King Chulalongkorn Memorial Hospital grant number GCE 59-009-30-005. The funders had no role in study design, data collection and analysis, decision to publish, or preparation of the manuscript.

==============================
Background

The COVID-19 virus has been an emerging disease causing global outbreaks for over a year. In Thailand, transmission may be controlled by strict measures that could positively and negatively impact physical health and suicidal behavior.

Methods

The incidence of COVID-19 was retrieved from the Department of Disease Control (DDC). The impact of viral diseases was retrieved from the open-source of the DDC and King Chulalongkorn Memorial Hospital. The road accidents data were from the Thai Ministry of Transport. The suicidal behavior data were obtained from the Department of Mental Health. We compared data from the year 2019 with the pandemic COVID-19 outbreak period in 2020, before lockdown, during lockdown, easing, and new wave period using unpaired t-test and least-squares linear regression. We compared the impact of the outbreak on various data records in 2020 with corresponding non-outbreak from 2019.

Results

There was a significant decline in cases of influenza (p < 0.001) and norovirus (p = 0.01). However, there was no significant difference in RSV cases (p = 0.17). There was a dramatic increase in attempt to suicides and suicides (p < 0.001). There was no impact on roadside accidents and outpatient department visits.

Discussion

The extensive intervention measures during lockdown during the first wave positively impacted total cases for each period for acute respiratory and gastrointestinal tract diseases, car accidents, and injuries and negatively impacted indicators of suicidal behavior. The data support government policies that would be effective against the next outbreak by promoting the “new normal” lifestyle.

Introduction

The coronavirus disease 2019 (COVID-19) is a recently emerging infectious disease. The first outbreak was found in Wuhan, Hubei, mainland China in December 2019. The World Health Organization (WHO) declared a Public Health Emergency of International Concern (PHEIC) on January, 30 2020 and a pandemic on March 11, 2020 (World Health Organization, 2020a; World Health Organization, 2020b).

Early 2020

The Thai Department of Disease Control (DDC) began screening direct flight passengers from Wuhan on January 3 by a Thermoscan device at the airport (Hinjoy et al., 2020) and sent a suspected case to quarantine. A traveler from Wuhan with fever on January 8, confirmed by RT-PCR on January 12 and with official reported SARS-CoV-2 infection on January 13 (Emergency Operation Center Department of Disease Control Thailand, 2020; World Health Organization, 2020a; World Health Organization, 2020b) had the first COVID-19 case reported outside mainland China (Okada et al., 2020). The first domestic case was found on January 31, a taxi driver who drove a foreign passenger (Pongpirul et al., 2020). Fifty confirmed cases of COVID-19 were reported from January 13–March 9, most of which were foreign travelers. In early 2020, a Thai group from a high-risk area was placed in isolation and under observation for 14 days called “State Quarantine” (SQ) (Limsawart et al., 2020; Department of Disease Control, 2020a; Department of Disease Control, 2020b; Department of Disease Control, 2020c; Department of Disease Control, 2020d) with Government subsidized the expense. The first confirmed SQ case occurred on February 8, in a Thai returning from Wuhan (Department of Disease Control, 2020a; Department of Disease Control, 2020b; Department of Disease Control, 2020c; Department of Disease Control, 2020d).

The Thai DDC activated the Emergency Operation Center (EOC) on January 4, then officially declared COVID-19 a dangerous communicable disease on February 29. So, the government used intensive measures to limit the outbreak through legislation (Limsawart et al., 2020).

During this period, hygiene was promoted through social distancing, hand sanitization, and wearing a facemask. Supplies of alcohol-based sanitizers and surgical masks, however, were insufficient. The Thai Department of Health recommended February 14 that people wear a fabric facemask (Bureau of Information Office of the Permanent Secretary, 2020). The government then allowed the alcohol industry to increase alcohol production for direct retail sanitizer sale (MGR Online, 2020).

Although Thailand responded well to an early outbreak, there was concern about transmission in two clusters: (1) entertainment venues and (2) the boxing stadium. The largest outbreak traced to the boxing stadium occurred on March 6, with over 4,500 participants from around the country. After these clusters, domestic cases increased enormously, with the highest peak of 188 patients on March 18, and daily new cases numbered over 100 per day on week 12–15. Participants returned to their hometowns after Bangkok’s lockdown on March 22 and thus spread the disease nationwide (Triukose et al., 2021).

Tightened measures

Nationwide outbreak situation led to the imposition of measures that could reduce the transmission as shown in the Fig. S1. In addition, the government suspended international flights and allowed only Thai and foreigners who worked or resided in Thailand to return with strict requirements: COVID-19-free medical certificate within 72 h before a flight, certificate of entry (COE), and fit-to-fly health certificate from the Thai Royal Embassy or Consulate before boarding, and health insurance for foreigners. They were detained in SQ for 14 days beginning April 4 (Limsawart et al., 2020; Department of Disease Control, 2020a; Department of Disease Control, 2020b; Office of the Permanent Secretary for Interior, 2020).

After these interventions, daily confirmed new cases substantially decreased. The last domestic case was reported on May 25. After that, there was no local transmission. The reported new COVID-19 cases were from the SQ only (Department of Disease Control, 2020a; Department of Disease Control, 2020b; Department of Disease Control, 2020c; Department of Disease Control, 2020d).

Easing

For successful disease control, the government implemented easing measures in five phases (Triukose et al., 2021; Department of Disease Control, 2020a; Department of Disease Control, 2020b; Department of Disease Control, 2020c; Department of Disease Control, 2020d).

The first easing on May 3 allowed reopening of fresh markets, food and beverage shops outside department stores, barbers, outdoor exercise facilities except for group activities, and pet shops, and canceled alcoholic beverage trade prohibitions.

The second easing on May 17 allowed reopening of shopping centers except for cinemas, bowling alleys, and amusement parks; hotel meeting rooms with limited participants; beauty clinics; only fitness centers outside department stores; indoor stadiums with three persons per one team; public parks; museums; learning centers; libraries; and galleries and shortened the curfew period (23:00-04:00).

The third easing on June 1 allowed group dining out with maintaining at least 1 meter social distance and full service for shopping centers, barbers, beauty clinics, spas, traditional massage parlors, tattoo parlors, body-piercing shops, fitness clubs, cinemas, exhibition venues, and child daycare centers if they followed the CCSA recommendations, and reopened zoos and allowed group sports for practice only and boxing without a trainer.

The fourth easing on June 15 allowed reopening tutorial schools with appropriate hygiene measures and distanced seating; serving alcohol in restaurants except for pubs, bars, and karaoke shops; seating passengers next to each other; reopening saunas; and allowing competition sports without spectators.

The fifth easing on July 1 allowed businesses such as pubs, bars, karaoke venues, gaming centers, internet cafes, and soapy massage parlors to reopen and close before midnight. Public vehicles were allowed to carry passengers at 70% of capacity because of schools reopening (Limsawart et al., 2020; Department of Disease Control, 2020a; Department of Disease Control, 2020b; Department of Disease Control, 2020c; Department of Disease Control, 2020d; Office of the Permanent Secretary for Interior, 2020).

The first wave results

Thailand is the few countries that completely controlled disease transmission in the first wave. Domestic cases no longer occurred by 100 days after the last case on May 25. A new domestic case occurred on September 4 of unknown origin. In this case, the patient worked in entertainment venues and was incarcerated in the penitentiary at the time the disease was detected. No close contacts of the patient had positive PCR results (Department of Disease Control, 2020a; Department of Disease Control, 2020b; Department of Disease Control, 2020c; Department of Disease Control, 2020d; Office of the Permanent Secretary for Interior, 2020).

In the last trimester of 2020, neighborhoods around Thailand confronted a high incidence of COVID-19, and a problem remained on borders because of smuggling immigrants entering Thailand to look for a job. Thailand dealt with this situation by concentrating border patrol police, armed forces patrols, and health volunteers on the border and focused checkpoints along the road in these areas. Illegal foreign immigrants were sent back to their country of origin. Thai workers illegally entering Thailand were sent to local quarantine, another type of SQ (Department of Disease Control, 2020a; Department of Disease Control, 2020b; Department of Disease Control, 2020c; Department of Disease Control, 2020d; Office of the Permanent Secretary for Interior, 2020).

Thailand has strengthened surveillance along borders. Some immigrants could otherwise enter Thailand without being arrested, leading to a new cluster. On November 26, the first confirmed case in this cluster occurred in a Thai worker in 1G1-7 hotel-cum-entertainment complex, Tachilek, Myanmar. After the first case was found on November 26, the number of new infections linked to the 1G1-7 hotel had risen to over 10 Thai workers who illegally entered to Thailand via natural borders. This cluster could spread the disease to two indigenous Thai (Department of Disease Control, 2020a; Department of Disease Control, 2020b; Department of Disease Control, 2020c; Department of Disease Control, 2020d; Office of the Permanent Secretary for Interior, 2020).

The new wave

Although Thailand could control previous clusters, smuggling immigrants into Thailand led to the new wave in December 2020. A new wave from the Central Shrimp Market, Samut Sakhon Province on December 17. This cluster was related to Myanmar workers who smuggle immigrants to Thailand, and they live in crowded conditions with poor hygiene. After this, active case finding in Myanmar workers found 1,390 cases until the end of 2020. Other clusters occurred at that time in big bike groups, entertainment venues, restaurants, and illegal gambling dens. The illegal gambling den cluster that occurred in Eastern Thailand was difficult to control because gamblers conceal their timeline information. This cluster had over 300 cases and remains untraceable to all gamblers. This new wave caused 1,110 confirmed cases in medical facilities and spread disease over 50 provinces within a few days (Department of Disease Control, 2020a; Department of Disease Control, 2020b; Department of Disease Control, 2020c; Department of Disease Control, 2020d; Office of the Permanent Secretary for Interior, 2020).

During the new wave, nationwide lockdown was not used but the country was classified into four zones: red, orange, yellow, and green according to the number of reported cases. The zone classification was announced on December 24 (Department of Disease Control, 2020a; Department of Disease Control, 2020b; Department of Disease Control, 2020c; Department of Disease Control, 2020d).

In the early wave, Samut Sakhon was defined as a red zone. Neighborhoods (Bangkok, Nakhon Pathom, Samut Songkram, Ratchaburi) and those with a high incidence were defined as orange zones. Provinces with sporadic cases were defined as yellow zones. Any other was defined as a green zone. However, red and orange zones were extended to many provinces, and other regions became yellow zones to counter this situation (Office of the Permanent Secretary for Interior, 2020).

Summary of cases

On December 31, 2020, there were 6,884 confirmed cases in Thailand with 61 deaths (mortality 0.89%). In contrast, worldwide confirmed cases exceeded 81 million with a total case fatality of approximately 1.8 million (mortality 2.21%) (Department of Disease Control, 2020; Center for Systems Science and Engineering [CSSE] at Johns Hopkins University [JHU], 2020). The outbreak and events in 2020 involved the pandemic shown in Fig. 1 and Table 1.

Figure 1 The COVID-19 outbreak in Thailand and related events.

Table 1 Timeline of the COVID-19 outbreak in Thailand.

	Date	Weeks	Event	Detail	
	January 4	1	Activated EOC	DDC activated the Emergency Operation Center (EOC)	
	January 13	3	First confirmed case	First report of COVID-19 case outside mainland China	
	January 22	4	EOC upscaled	Thai Prime Minister promoted the EOC to be the Ministerial level	
	January 31	5	First domestic case	First report of COVID-19 in taxi driver who had close contact with a foreign passenger	
	February 4	6	Group from Wuhan	138 Thai nationals return from Wuhan and are sent to state quarantine. This is the first state quarantine measure that was compulsory for people coming from an at-risk country	
Before Lockdown (week 1–11)	February 8	6	The first case in state quarantine	The first case in state quarantine in Thai nationals returning from Wuhan	
	February 29	9	Declared as a dangerous communicable disease	Ministry of Public Health announced COVID-19 as a dangerous communicable disease by the Communicable Diseases Act, B.E. 2558 (A.D. 2015)	
	March 1	10	First fatal case	The first fatal case, in a patient who had an underlying disease	
	March 3	10	Group from South Korea	Thai workers returned from South Korea	
	March 9	10	Entertainment venues cluster	Confirmed cases from entertainment venues	
	March 11	11	Boxing stadium cluster	Confirmed cases from entertainment venues	
	March 15	12	Group from Dakwah Pilgrimage	132 Thai nationals from Dakwah Pilgrimage	
	March 16	12	New Year revocation	Thai New Year (Songkran festival) on April 13–15 canceled	
	March 18	12	Social distance	Social distancing measures initiated	
	March 21	12	Schools and universities closure		
	March 22	12	Bangkok lockdown	Bangkok Governor announces lockdown measures	
	March 26	13	Thailand emergency decree	Emergency decree included the closure of borders	
	March 26	13	Inception of the CCSA	Thai Government establishes the Center for COVID-19 Situation Administration (CCSA)	
Lockdown (week 12–18)	April 3	14	Nationwide curfew	Nationwide curfew (22:00–04:00) for all except medical personnel, law enforcement, logistics workers, or those who have a necessary reason to be out at that time	
	April 4	14	Enforcement of state quarantine	Enforcement of state quarantine to all arriving international flight without exception	
	April 12	16	Alcoholic beverage trade prohibition		
	April 13–15	16	New Year celebration canceled		
	May 3	19	First phase easing		
	May 17	21	Second phase easing		
	June 1	23	Third phase easing		
Easing (week 19–50)	June 15	25	Fourth phase easing		
	July 1	27	Fifth phase easing		
	July 25–28	31	1st New Year rescheduling	1st Thai New Year rescheduling on July 25–28, 2020	
	September 4–7	36	2nd New Year rescheduling	2nd Thai New Year rescheduling on Sep 4–7, 2020	
	December 18	51	The first case related to the Central Shrimp Market	The first case in the shrimp market cluster	
New wave (week 51–53)	December 19	51	Samut Sakhon lockdown	Thai DDC announce lockdown measures with a curfew (22:00–05:00) including travel restriction only for foreign workers	
	December 24	52	The first case in an illegal gambling den cluster	The first case in an illegal gambling den cluster	
	December 24	52	Zone classification	Zone classification by colors (red, orange, yellow, and green), depending on the local situation	

Previous studies revealed that the COVID-19 pandemic was associated with a decrease in incidence of respiratory tract and gastrointestinal tract infections (Lee & Lin, 2020; Suntronwong et al., 2020; Wong, Leung & Lee, 2020). However, these studies were conducted during the pre- and early-pandemic period (January–early April 2020). Long-term data on the dynamic changes of incidence rates were limited, especially when different measurements for controlling the COVID-19 were implemented.

Our study aimed to reveal the impact of COVID-19 control measures on incidence of other viral infections and effects on suicidal behavior, accidents, and outpatient department (OPD) visits between 2019 and 2020. Positive and negative effects were compared with the previous data before the outbreak. The results could help health policymakers prepare for measures to reduce adverse events from interventions to contain a contagious disease.

Materials & Methods

We assessed and compared the data records on acute viral respiratory tract and gastrointestinal tract infections, OPD visits, road accidents, and suicidal behavior in Thailand during the COVID-19 period vs. the analogous period in 2019. This study used secondary anonymous information and did not require ethical approval.

Data sources

This study used Government agency statistics. The COVID-19 data were retrieved from the Thai DDC (Department of Disease Control, 2020a; Department of Disease Control, 2020b; Department of Disease Control, 2020c; Department of Disease Control, 2020d). Data on the incidence of respiratory tract and gastrointestinal infections were from clinical records from the hospital in Bangkok (Suntronwong et al., 2020). Data on the incidence of vector-borne infection were from Report 506 (Bureau of Epidemiology, 2021). OPD visits data were retrieved directly from King Chulalongkorn Memorial Hospital (KCMH). Suicidal behavior (attempt to suicide and suicide) data were retrieved from the self-harm surveillance system (Report 506S) of the Thai Department of Mental Health (Department of Mental Health MoPH Thailand, 2021). The accident data were retrieved from the TRansport Accident Management Systems (TRAMS) report system of the Thai Ministry of Transport (Ministry of Transport, Thailand, 2021). Data were from the period January 1 to December 31, 2020. Control data were those before COVID-19 emerged and compared an event, holiday, and festival for the same time period in 2019. The unpaired t-test and least-squares linear regression were used to compare total cases each year and each period (Before lockdown (week 1–11), Lockdown (week 12–18), Easing (week 19–50), and New wave (week 51–53)). Bonferroni correction was used to corrected p value thresholds in each period, p value < 0.05 for total cases in each year, and <0.0125 for each period were considered statistically significant.

Results

This study provides data about other viral diseases, suicidal behavior, OPD visits, and traffic situations in the COVID-19 era.

Impact on other viral infections

There was a significant reduction in total influenza cases in 2020 as compared to 2019 (Fig. 2A) as shown in Table 2. There was a positive impact on reduction of influenza cases in 2020 during lockdown and easing. The total number of cases of respiratory syncytial virus (RSV) in 2019 compared to 2020 (Fig. 2B) was not significantly different, as shown in Table 2.

Figure 2 Relationship of COVID-19 cases to other viral diseases, 2019–2020.

(A) Influenza, (B) respiratory syncytial virus (RSV), (C) rotavirus, (D) norovirus, and (E) dengue.

Table 2 Impact of COVID-19 pandemic on other viral infections in Thailand, 2019-2020.

		n (%)a	Trend (per week)	
		2019	2020	p value	2019	2020	
Influenzab	Total	1,230 (25.8)	303 (12.3)	<0.001	−0.1415	−0.4884	
	Before lockdown	377 (33.7)	287 (22.2)	0.29	1.927	−3.318	
	Lockdown	182 (77.1)	0 (0.0)	0.001	−1.357	0.0	
	Easing	621 (61.9)	10 (0.4)	<0.001	0.838	0.04032	
	New wave	50 (49.0)	6 (2.9)	0.07	−5.000	0.0	
RSVb	Total	171 (3.6)	297 (12.1)	0.17	0.1035	0.3743	
	Before lockdown	12 (0.9)	8 (0.7)	0.39	0.009091	−0.2000	
	Lockdown	6 (1.2)	1 (0.4)	0.03	0.1429	−0.07143	
	Easing	147 (5.4)	256 (9.3)	0.14	0.1292	0.9479	
	New wave	6 (2.9)	14 (6.8)	0.47	−1.000	−5.000	
Rotavirusb	Total	25 (6.0)	164 (37.3)	<0.001	−0.01895	−0.2857	
	Before lockdown	15 (12.3)	158 (39.2)	<0.001	−0.4909	−1.336	
	Lockdown	1 (1.2)	6 (17.6)	0.27	0.03571	−0.2143	
	Easing	3 (1.6)	0 (0.0)	0.08	−0.01118	0.0	
	New wave	6 (30.0)	0 (0.0)	0.12	1.500	0.0	
Norovirusb	Total	138 (14.2)	59 (10.3)	0.01	0.07273	−0.09539	
	Before lockdown	28 (23.0)	53 (13.0)	0.17	−0.4182	−0.5091	
	Lockdown	4 (4.7)	3 (8.1)	0.79	−0.3214	0.1071	
	Easing	79 (11.9)	3 (2.3)	<0.001	0.1219	−0.001283	
	New wave	27 (26.7)	0 (0.0)	0.049	−3.500	0.0	
Denguec	Total	131,157	71,292	<0.001	35.86	12.35	
	Before lockdown	12,545	7,134	<0.001	39.41	1.391	
	Lockdown	8,188	4,804	0.001	55.36	86.61	
	Easing	105,975	58,883	<0.001	−38.86	−50.98	
	New wave	4,449	471	0.14	250.0	−307.0	
Notes.

a Percentage of positive cases from all specimens.

b Viral infection data were obtained from clinical data from the hospital in Bangkok (Suntronwong et al., 2020).

c Dengue data were from Report 506, Thai DDC (Bureau of Epidemiology, 2021).

Total cases of rotavirus were significantly higher in 2020 compared to 2019 (Fig. 2C), as shown in Table 2, but rotavirus cases in 2020 declined after lockdown. After an initial spike in early 2020, norovirus total cases in 2020 were significantly less compared to 2019 (Fig. 2D), as shown in Table 2. There was a positive impact on reducing norovirus cases in 2020 during the easing and new-wave periods.

Dengue total cases in 2020 compared to 2019 (Fig. 2E) were significantly reduced, as shown in Table 2. However, data in 2020 during the new wave period were not available for analysis.

Impact on outpatient department

There were no significant differences in patient visits to the OPD service in the KCMH in 2019 compared to 2020 (Fig. 3), as shown in Table 3. Although overall patient visits were not significantly different, there was a declining trend during the lockdown.

Figure 3 Patients visits to the OPD in the KCMH, 2019 and 2020.

Table 3 Impact of the COVID-19 pandemic on the Outpatient Department (OPD) in KCMH, 2019 and 2020.

		n	Trend (per week)	
		2019	2020	p value	2019	2020	
OPD service	Total	1,389,783	1,327,606	0.29	16.81	55.88	
	Before lockdown	283,013	300,562	0.52	651.3	819.8	
	Lockdown	176,443	136,424	0.09	35.64	−1440.0	
	Easing	868,961	812,904	0.10	64.50	208.1	
	New wave	61,366	77,716	0.66	−15429.0	−7704.0	
Notes.

n, number of outpatient clinic visits in KCMH.

Impact on suicidal behavior

The suicidal behavior data focused on two outcomes, attempt to suicides and suicides. Attempt to suicide cases significantly increased in 2020 compared to 2019 (Fig. 4A) as shown in Table 4. Suicide total cases in 2020 significantly increased compared to 2019 (Fig. 4B) as shown in Table 4. Attempt to suicides and suicides increased in 2020 before and during lockdown and decreased during easing.

Figure 4 Suicidal behavior.

Data from the self-harm surveillance system (Report 506S), Thai Department of Mental Health (Department of Mental Health, 2021), shows both (A) attempt to suicide and (B) suicide.

Table 4 Attempt to suicide and suicide, 2019–2020.

		n	Trend (per week)	
		2019	2020	p value	2019	2020	
Attempt to suicides	Total	8,098	10,006	0.001	2.331	−2.329	
	Before lockdown	1,418	2,651	<0.001	1.100	3.318	
	Lockdown	896	1,381	<0.001	−2.643	−4.750	
	Easing	5,230	5,661	0.35	6.049	−2.599	
	New wave	554	313	0.24	−91.50	−35.00	
Suicides	Total	2,520	4,022	<0.001	1.042	−1.145	
	Before lockdown	408	1,072	<0.001	0.1636	1.009	
	Lockdown	245	629	<0.001	−0.8214	−2.536	
	Easing	1,653	2,189	<0.01	2.476	−1.420	
	New wave	214	132	0.25	−31.50	−10.00	
Notes.

n, number of cases reported to the Department of Mental Health.

Impact on road accident incidences and fatalities

Indicators of the road accidents in Thailand were car accidents, injuries from road accidents, and fatalities from road accidents from TRAMS data in 2019 compared to 2020. Road accidents (Fig. 5A), injuries from road accidents (Fig. 5B), and fatalities from road accidents (Fig. 5C) were not significantly different during the 2020, as shown in Table 5. These cases tended to increase in 2020 during easing but declined somewhat during lockdown.

Figure 5 Road accident incidences and fatalities, 2019-2020.

Data from TRAMS report system, the Thai Ministry of Transport (Ministry of Transport, Thailand, 2021) (A) road accidents (B) injuries from road accidents, and (C) fatalities from road accidents.

Table 5 Pandemic impact on traffic indicators, 2019–2020.

		n	Trend (per week)	
		2019	2020	p value	2019	2020	
Road accidents	Total	19,575	21,587	0.17	0.2000	1.998	
	Before lockdown	4,006	4,607	0.14	−14.09	−3.309	
	Lockdown	3,560	2,356	0.17	15.18	−5.571	
	Easing	9,931	12,840	<0.001	1.264	3.269	
	New wave	2,078	1,784	0.73	285.5	251.5	
Injuries from road accidents	Total	16,605	16,585	0.99	0.2272	1.592	
	Before lockdown	3,287	3,934	0.21	−15.55	−11.55	
	Lockdown	3,396	1,578	0.08	21.07	−5.679	
	Easing	7,821	9,455	0.02	0.6420	3.987	
	New wave	2,101	1,618	0.63	392.0	239.0	
Fatalities from road accidents	Total	2,862	3,137	0.24	0.1719	0.3070	
	Before lockdown	578	715	0.03	−3.191	−0.2273	
	Lockdown	537	304	0.10	3.786	−1.893	
	Easing	1,464	1,846	<0.001	1.058	0.6620	
	New wave	283	272	0.93	9.000	49.00	
Notes.

n, number of cases reported to the Ministry of Transport.

Discussion

The COVID-19 pandemic and interventions have led to changes in human lifestyles with people becoming accustomed to a “new normal”. This study focused on the impact in Thailand on various aspects, including public health, suicidal behavior, socioeconomics, and road accidents. Both positive and negative effects were found when comparing pandemic and pre-pandemic periods.

Because of improved personal hygiene, including hand washing and alcohol-based sanitization (Gupta & Lipner, 2020), facemask use (Liang et al., 2020), and social distancing, viral upper respiratory tract and viral gastrointestinal tract infections significantly decreased. Although rotavirus significantly increased prior to lockdown during early 2020, there were only six cases during the lockdown, and no more after that.

A vector-borne viral disease, dengue, was significantly reduced in this analysis as a result of the pandemic. National data surveillance (Bureau of Epidemiology, 2021) revealed that dengue has been outbreaking out every 2–3 years. The year 2020 was not an outbreak year in Thailand’s outbreak pattern. Of communicable diseases generally spread through the community, especially in the urban or crowded areas, the interventions focusing on a lockdown and travel restrictions can limit disease spread.

As COVID-19 spreads, the total number of patients in the OPD decreased during the nationwide outbreak in many hospitals, including KCMH. Some hospitals limited the number of daily patients and required appointments for non-emergency cases. Moreover, an online service was used to reduce the number of outpatients in hospitals by drug deliveries and telemedicine consultation. Patients also had access to a small clinic instead of the hospital to disease avoidance.

Mental health during the pandemic is as significant a problem as physical health. Various factors associated with mental health included misinformation, lack of knowledge, socioeconomic problems, social media use, and isolation/quarantine issues. The systematic review showed significantly increased depression, fear, anxiety, stress, suicidal behavior, job loss, financial crisis, nutrition disorders and increase work load in medical personnel (BC et al., 2021; Gao et al., 2020; Goodwin et al., 2020; John et al., 2020; Salari et al., 2020). These factors could lead to self-harm or suicidal behavior. The trend of suicide during the early month of the COVID-19 pandemic increase (Pirkis et al., 2021), however, each country has different situation, measurement, people attitude which could affect mental health and suicidal behavior. In Thailand 2020, there was an increase in suicidal behavior including attempts to suicides and suicides.

Certain populations are a higher risk. Continuing treatment for people with pre-existing mental health problems is challenging because of lockdown and travel restrictions. Health care workers are also vulnerable because of increased workload and concern about disease exposure (Nochaiwong et al., 2020). People in SQ have unpleasant experiences being isolated for 2 weeks. Unemployed people suffered from financial and socioeconomic distress (Blakely, Collings & Atkinson, 2003). More importantly, people who were cured of COVID-19 have been stigmatized by their community (Roberto, Johnson & Rauhaus, 2020). There are additional deteriorating economics. The gross domestic product (GDP) in 2020 decreased by 7.1% (in 2019 it increased by 2.4%) (International Monetary Fund. Thailand, 2020).

Generally, Thailand celebrates the Thai New Year (Songkran festival) during the middle of April (April 13–15). In the past several years, the highest number of accidents occurred during this period due to the massive transport out of Bangkok. In 2020, injuries and fatalities from road accidents dramatically decreased during the Songkran festival. The reduction was due to the fact that the Government rescheduled the Songkran festival to July 25–28. Drunk driving has been a major cause of road accidents (Ministry of Transport, Thailand, 2021). In the middle of April 2020, alcohol trade were prohibited, thus resulting in a reduction of drunk driving and road accidents.

However, the total numbers of road accidents, injuries, and fatalities increased significantly in 2020. During the first easing phase, measures against the outbreak were not strict. People were free to travel without restriction and alcohol trade was permitted, so there were no differences in road accidents in 2020 and 2019.

Limitations of this study included the use of secondary data that did not provide complete information such as age, gender, or other information. Subjects could not be stratified into subgroups for further analysis. Only confirmed cases in the medical facilities were reported to the central MoPH agency. Data from other viral infections (Influenza, RSV, rotavirus, and norovirus) and outpatient department came from only some hospitals around Bangkok because national data on RSV and norovirus are not available, and national data on influenza are from suspected and confirmed cases. Furthermore, lockdown measure could be a confounding variable for the presentation of all events.

Conclusions

This study assessed the effects of interventions involving the COVID-19 pandemic in Thailand in 2020. Strict measures against the first wave could control the outbreak with substantially reduce morbidity and mortality within a few weeks. Disease impact can also be lessened by adopting a “new normal” lifestyle during the COVID-19 era. However, measures imposed during the first wave led to devastating economic consequences and negative effects on suicidal behavior. This measures during the lockdown of the first wave positively impacted total cases for each period for acute respiratory and gastrointestinal tract diseases, road accidents and fatalities.

Supplemental Information

Supplemental Information 1 Timeline of events during the first wave and easing period of COVID-19 outbreak in Thailand (in 2020)

Click here for additional data file.

Supplemental Information 2 Raw data

The COVID-19 timelines in the year 2020, other viral infections (included dengue infection), suicidal behavior, and road accident incidences and fatalities.

Click here for additional data file.

We are grateful to the staff of the Center of Excellence in Clinical Virology, Chulalongkorn University, and King Chulalongkorn Memorial Hospital, and the Thai Red Cross Society for their technical and administrative assistance. We would like to acknowledge the Department of Information Technology, King Chulalongkorn Memorial Hospital, the Thai Red Cross Society for provided the data from the hospital. We are appreciative and grateful to the staff and nurses of the Institute of Urban Disease Control and Prevention (IUDC), Department of Disease Control, Ministry of Public Health, Thailand for the support and valuable data.

Additional Information and Declarations

Competing Interests

Author Contributions

Data Availability

The authors declare there are no competing interests.

Ritthideach Yorsaeng conceived and designed the experiments, performed the experiments, analyzed the data, prepared figures and/or tables, authored or reviewed drafts of the paper, and approved the final draft.

Nungruthai Suntronwong, Ilada Thongpan, Watchaporn Chuchaona, Fajar Budi Lestari, Siripat Pasittungkul and Jiratchaya Puenpa performed the experiments, prepared figures and/or tables, and approved the final draft.

Kamolthip Atsawawaranunt and Chollasap Sharma analyzed the data, prepared figures and/or tables, and approved the final draft.

Natthinee Sudhinaraset and Yong Poovorawan conceived and designed the experiments, analyzed the data, authored or reviewed drafts of the paper, and approved the final draft.

Anek Mungaomklang, Rungrueng Kitphati and Nasamon Wanlapakorn analyzed the data, authored or reviewed drafts of the paper, and approved the final draft.

The following information was supplied regarding data availability:

The raw data is available in the Supplementary File.

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
