# Peer review of "The impact of COVID-19 and control measures on public health in Thailand, 2020"

_PeerJ, doi:10.7717/peerj.12960_

## Round 0.1 · original submission · Major Revisions

Kindly mention the source of the secondary data and how validity and reliability was measured.

Reviewer 1 ·

Basic reporting

It is a nicely written paper with a novel research question. The discussed literature is time-bound and adequate.

Experimental design

Methods are well mentioned.

Validity of the findings

As the authors used already available secondary data from the other sources the quality could be compromised which has been mentioned in the limitations.

Additional comments

1. As the authors only assessed the suicidal attempts and suicides, I request to revamp the whole draft replacing mental health with suicidal behavior.
2. Authors should mention that lockdown could be a confounding variable for the presentation of all events, in the limitation section.
3. While discussing the suicidal behavior, authors are requested to discuss the below-mentioned papers
https://www.ncbi.nlm.nih.gov/pmc/articles/PMC7871358/
https://pubmed.ncbi.nlm.nih.gov/33862016/

·

Basic reporting

The article provides detailed insight into the background and timeline of COVID-19 pandemic in Thailand and the early measures taken to prevent the outbreak from getting worse. However, I would like to recommend a few suggestions to make the article better in writing and understanding.
1. Check for grammar and addition of articles that modify the nouns such as "THE Coronavirus disease" in sentence 49. Also, check for sentences where lesser words can be used to communicate a fact e.g. sentence 60-61 could be re-written as "50 confirmed cases of COVID-19 were reported from Jan 13- March 9, most of which were foreign travellers". I would recommend a grammar check.
2. According to Tantrakarnapa & Bhopdhornangkul(2020), in their paper called "Challenging the spread of COVID-19 in Thailand", the first case was reported in Thailand on Jan 22, and according to Mahikul et al., (2021) in "Evaluating the Impact of Intervention Strategies on the First Wave and Predicting the Second Wave of COVID-19 in Thailand", Thailand was the first country after China to report a case of COVID-19 on January 12th. So I would recommend checking the timeline again to confirm the dates.
3. the introduction gives a wide account of timeline of COVID-19 in Thailand and start of interventions. The dates are elaborate with a lot of detail which is commendable. However I feel sentence 72-75 should come before 66-70. It would give the timeline more flow.
4. Generally the whole article needs more consistency and flow to it and should have more words like "moreover", "furthermore" etc. to link one paragraph to another instead to giving a separate bullet point for each date and event.
5. the timeline is too long and exaggerated and I would recommend crunching it down to numbers and trends rather than whole long paragraphs.

Experimental design

1.The sources used for data collection, while being the correct go to source, might not represent the whole country sample. e.g. the data collected for suicides and OPD admissions has been gathered from a hospital in Bangkok. This sample does not represent both urban and rural areas.
2. The data collection has been done from government databases which will only represent the reported cases and thus cannot be generalised to people who were asymptomatic to diseases or did not report the disease in hospitals or to the government.
3. The diseases in the categories of viral diseases and vector borne diseases have no background context before methodology and up until discussion part of the paper its not clear which diseases the author is talking about. e.g. The paper mentions respiratory and Gastrointestinal infections and only in discussion does the author mention "viral gastritis". Thus the mention and context of viral and vector borne diseases seems vague.
4. Mental health is measured just by suicides and attempt to suicide while the author does mention other mental health issues such as anxiety, depression and isolation due to quarantine to be more widespread during pandemic. This could be expanded upon more in the current paper.
5. The road accidents seems like an irrelevant variable to public health measures of COVID-19 pandemic,

Validity of the findings

The findings are derived from a reliable government source and seem accurate.
The conclusion and results of the study are well documented in terms of graphs, data and trends.

Additional comments

The article provides a deep insight into the COVID-19 conditions in Thailand and the measures taken. However, other than early intervention, other factors use are the same as other countries worldwide. The paper could have used a more comparative approach with other countries and focused more on aspects such as contact tracing, technological and vaccination interventions.
This would help to make the paper more informative.

---

## Round 0.2 · accepted · Accept

Thanks for making all the necessary changes.

Reviewer 1 ·

Basic reporting

Nicely written paper

Experimental design

Methods are rigorous

Validity of the findings

Data are statistically sound

Additional comments

Thanks for addressing commets.

·

Basic reporting

The changes made by the author are satisfactory. The grammar and flow of the article are better and well written.

Experimental design

The experimental design and data collection is reported much better and is satisfactory.

Validity of the findings

The findings reported are valid and satisfactory.

Additional comments

All the changes made by the author according to recommendations are enough and satisfactory.